# Healthy Sleep Every Day Keeps the Doctor Away

**DOI:** 10.3390/ijerph191710740

**Published:** 2022-08-29

**Authors:** Cailan Lindsay Feingold, Abbas Smiley

**Affiliations:** Westchester Medical Center, New York Medical College, New York, NY 10595, USA

**Keywords:** sleep, chronic disease, mechanism

## Abstract

When one considers the big picture of their health, sufficient sleep may often go overlooked as a keystone element in this picture. Insufficient sleep in either quality or duration is a growing problem for our modern society. It is essential to look at what this means for our health because insufficient sleep increases our risks of innumerable lifechanging diseases. Beyond increasing the risk of developing these diseases, it also makes the symptoms and pathogenesis of many diseases worse. Additionally, consistent quality sleep can not only improve our physical health but has also been shown to improve mental health and overall quality of life. Substandard sleep health could be a root cause for numerous issues individuals may be facing in their lives. It is essential that physicians take the time to learn about how to educate their patients on sleep health and try to work with them on an individual level to help motivate lifestyle changes. Facilitating access to sleep education for their patients is one way in which physicians can help provide patients with the tools to improve their sleep health. Throughout this paper, we will review the mechanisms behind the relationship between insufficient sleep health and chronic disease and what the science says about how inadequate sleep health negatively impacts the overall health and the quality of our lives. We will also explain the lifechanging effects of sufficient sleep and how we can help patients get there.

## 1. Introduction

A good night’s sleep is often the first thing sacrificed as our schedules fill up with commitments. Most people are too quick to lose a couple of hours of sleep, without knowing the damaging effects it can have on their bodies. The truth is, sleep is a keystone piece in the bigger picture of our health, and it has widespread effects on our bodies. The American adult population has experienced a large decline in sleep health. For the purposes of this paper, sleep health is used to account for multiple sleep characteristics including duration and quality. Sleep insufficiency denotes sleep health problems involving either sleep quality, sleep quantity or both. As sleep quality and duration is falling, rates of chronic diseases are simultaneously rising. This begs the question: what is the effect of sleep debt on chronic health problems?

The picture of sleep in America has changed drastically over time. National Sleep Foundation guidelines recommend that healthy adults aim for 7 to 9 h of sleep each night. According to previous data, our society was much closer to this goal back in the mid-twentieth century than we are now. A 1959–1960 American Cancer Society Survey found that 2% of their sample slept less than 6 h per night (defined as short sleep). Today, more than 30% of the adult population sleeps less than 6 h per night [1]. In 1910, it is estimated that Americans slept for 9 h each night on average, but today Americans sleep less than 7.5 h on average [2,3]. Many factors account for this shift, but one essential influence is the modern 24/7 access to the world through our smartphones, making it easier to procrastinate disconnecting and shutting down.

As our time asleep has fallen over the twentieth and twenty-first centuries, obesity has been concurrently rising. Obesity is a major and urgent public health problem in America that has only worsened over time. The portion of the adult population with a BMI of at least 30.0 (standard for obesity) has climbed from less than 15% in the early 1960s to 42.4% in 2018 [4,5]. Today that number is likely close to 45%, indicating obesity rates have increased by approximately 200% in the last 60 years. There are four factors that are contributing to this drastic increase: diet/nutrition, sedentary behavior/inactivity, depression/anxiety/stress, and perhaps most importantly, sleep.

In addition to a concurrent rise in the prevalence of obesity, depression, anxiety, and stress are also on the rise within the adult population. A positive relationship between insufficient sleep and mental health issues has been shown to exist [6,7]. Sleep insufficiency has been demonstrated to affect many other chronic conditions we will discuss here as well, including hypertension, metabolic syndrome, etc.

Substandard sleep health is contributing to many of the major public health crises that America and the world faces and is often overlooked as a major factor. Importantly, sleep is one of the easier factors to change, and thus it is essential to consider how the impact of sleep debt is advancing chronic illness. Lack of sleep, in either quality or quantity, impacts the other three factors (diet, sedentary behavior, depression/anxiety/stress) that are accountable for the increase in obesity, and therefore, sleep health impacts chronic disorders both directly and indirectly. Changing ones’ sleep schedule is a less taxing goal for individuals looking to improve their short- and long-term health. Here, we have compiled recent findings from the existing literature and made connections between data and conclusions from different studies in order to tell a detailed and comprehensive story of the impact of sleep health on our lives. In the paragraphs to come, we will explore together the complex relationship of sleep both with chronic disease, and with the other factors influencing chronic disease to better understand how sleep health affects our lives.

## 2. Inadequate Sleep Health and Obesity

### 2.1. Trends in Sleep Duration, Obesity, and Dietary Behaviors over Time

To decipher whether the changing trends in sleep duration and chronic conditions are related to one another, it is necessary to look at the data side by side.

The data represented in Figure 1 illustrates how these trends have changed over the last 60 years in the United States. Importantly, it accounts not only for sleep duration and the prevalence of obesity, but also includes an additional element, the number of McDonald’s restaurants, which is representative of behavioral changes over time. Wang et al., reviewed a number of studies that demonstrated a positive correlation between fast food chain growth and obesity [8]. In order to support the growth of the fast-food chain over this time period, a growing number of customers had to exist. Thus, there is a potential mechanism by which sleep insufficiency may be promoting behavioral changes with regard to what kind of food people choose to consume. The growth in Mcdonalds’ restaurants is a direct index of fast-food consumption and an indirect index of food behavior. Food behavior is influenced by many factors including but not limited to sleep health. As the percentage of American adults sleeping for less than 6 h per night increases, the percentage of obese adults does too.

This compilation of data suggests a remarkable relationship between the changes in sleep duration and the rise in the prevalence of obesity. To determine how to manage the effects of sleep on human health, we must now consider the physiological and behavioral mechanisms by which short sleep is increasing obesity and other chronic diseases among American adults.

### 2.2. A Closer Look at the Relationship between Sleep Health and Obesity

Numerous studies have demonstrated a relationship between short sleep duration and increased risks of obesity [9,10,11]. Obesity is one of the largest public health crises facing American society, and it is a serious condition that has an established role in predisposing individuals to several chronic diseases. Obesity is also extremely costly; between 2001 and 2016, the aggregate medical costs of obesity more than doubled to $260.6 billion in the United States [12]. There are many aspects of modern society that play a role in this epidemic, and sleep insufficiency is no exception. For this reason, it is necessary to understand how inadequate sleep health is contributing to the prevalence of obesity.

One of the ways in which sleep loss contributes to obesity is in its ability to alter the behavior and choices of individuals. Short sleep is associated with increased feelings of hunger and reduced satiety [13,14]. Adults subject to insufficient sleep in laboratory controlled conditions demonstrated an increase in snacking, especially at night, as well as a preference for foods higher in carbohydrates [15]. This finding was corroborated by a study with non-laboratory controlled conditions as well [16]. Sleep deprived adults are also willing to spend more money on food than their well-rested counterparts [13]. Taken together, evidence suggests that insufficient sleep drives individuals to make less healthy choices when it comes to their dietary habits.

In a similar fashion, short sleep also increased the likelihood of sedentary behavior and inactivity [17,18]. This may be partially due to increased feelings of fatigue following nights of decreased sleep. With sleep insufficiency increasing the likelihood of poor dietary choices and less active lifestyles, it is clear why individuals who suffer from a lack of sleep are in a prime position to gain weight. Importantly, there is solid data to back up this claim. The Nurse Health Study, which looked at a sample of close to 70,000 women, found that women who were sleeping less than 5 h per night were 32% more likely to experience major weight gain (defined as 15 kg or more) over the 16 year duration of the study [19].

With the evidence supporting a role for insufficient sleep in behavioral changes contributing to weight gain, it is important to now consider the mechanisms by which sleep loss mediates these behaviors. There are several endocrine changes following sleep deprivation that may play a role in influencing behaviors that disrupt the energy balance by increasing energy intake while decreasing energy expenditure.

Leptin and ghrelin are two hormones known to regulate feelings of hunger and satiety. Many studies have demonstrated a relationship between short sleep and changing levels of these hormones. Ghrelin, the hormone that promotes hunger, is elevated following short sleep and total sleep deprivation [20,21]. In contrast, leptin aids in regulation of energy balance by suppressing hunger. Following a reduction in sleep duration, leptin levels are reduced [21,22,23]. Similarly, peptide YY, a gut hormone peptide, is secreted in response to meal consumption and acts as an appetite suppressant [24]. Short sleep duration was shown to be associated with significantly reduced levels of fasting peptide YY and feelings of fullness [25]. The effects of leptin, peptide YY, and ghrelin on appetite regulation, and their fluctuations following sleep deprivation are notable in consideration of the relationship between sleep and obesity. These changes in hormone levels are influencing feelings of increased hunger in sleep deprived individuals and may be driving decision making about food choices.

Another key hormone influenced by sleep loss is insulin. Research has demonstrated that substandard sleep health may be affecting appropriate insulin secretion through its effect on glucagon-like peptide 1 (GLP-1). GLP-1 is secreted following food consumption and promotes insulin secretion in response [26]. Importantly, it is also known to promote feelings of satiety, similar to leptin [27]. Benedict et al. has shown that acute sleep deprivation delays the GLP-1 response to breakfast in healthy subjects [28]. Therefore, sleep insufficiency not only looks to impact the insulin response to food consumption by delaying GLP-1, it may also contribute to delayed satiety that promotes eating more than necessary during a meal.

Studies have also shown that groups subjected to sleep deprivation demonstrated higher levels of insulin resistance compared with control groups [9,29,30,31]. Sleep deprivation has been shown to affect both hepatic and peripheral insulin sensitivity, with subjects demonstrating both an increase in glucose production and a decrease in glucose uptake [30,32]. Interestingly, insulin sensitivity is recovered following only two nights of full sleep after a bout of acute sleep deprivation [33,34]. Together these findings suggest a relationship exists between insulin sensitivity and sleep deprivation at both the hepatic and peripheral levels.

This reduction in insulin sensitivity following sleep loss might play a role in the reduced leptin levels of sleep deprived individuals. The production of leptin by adipocytes is a process that is stimulated by insulin signaling. However, in a state of increased insulin resistance, adipocytes will be less responsive to insulin, thus producing and releasing less leptin as a result. Some evidence for this relationship does exist; studies have demonstrated that insulin helps to increase transcription of leptin mRNA [35,36]. Additionally, there is evidence that insulin not only contributes to leptin levels at the point of transcription, but also in the exocytosis of leptin from adipocytes [37]. Importantly, adipocytes are not an exception from the impacts of sleep loss on insulin resistance; the insulin signaling pathway in adipocytes is less active following sleep restriction [29]. Thus, in short sleep conditions, adipocytes are less responsive to insulin and consequently they likely produce and release less leptin.

Insulin sensitivity likely also has an influence on increased ghrelin levels. There are some studies that demonstrate a reciprocal relationship between ghrelin and insulin: before a meal, insulin is low and ghrelin is high, but following a meal insulin is high and ghrelin is low [38]. Even in fasting conditions the relationship between ghrelin and insulin appears to be inverse [39]. Therefore, if cells are less responsive to insulin, gastric cells might produce an inappropriate amount of ghrelin.

A key question here seems to be how short sleep duration is influencing insulin sensitivity, as insulin insensitivity appears to be upstream to the effects on leptin and ghrelin levels. The inflammatory effect of sleep deprivation might play a role. Short sleep duration is associated with an increase in a number of cytokines and inflammatory signaling molecules, but TNFα is one that is especially important when considering insulin signaling [40,41]. TNFα is significant in a conversation about obesity because it is produced and secreted by adipose tissue; studies have shown that in obese mice TNFα is overexpressed [42,43,44]. Importantly, there is a large amount of evidence for a role for TNFα in insulin resistance [42,43]. In one study, TNFα-deficient obese mice demonstrated significantly improved insulin sensitivity compared with mice containing a functional TNFα gene [45].

Further research has begun to uncover a mechanism for TNFα’s role in insulin resistance. Evidence demonstrates that TNFα impairs the uptake of glucose by preventing phosphorylation of a necessary step in insulin’s signaling of GLUT4 to the plasma membrane [46]. Therefore, high TNFα levels will cause peripheral cells to uptake less glucose from the blood in response to insulin signaling. Taken together, this suggests that the changes in leptin and ghrelin levels that are likely contributing to changes in dietary behavior stem from the effects of elevated TNFα on insulin sensitivity (insufficient sleep → elevated TNFα → insulin insensitivity → leptin & ghrelin changes → change in dietary behavior, Figure 2).

Considering the evidence for the role of cytokine TNFα in insulin resistance in obese models, it might be possible that the elevated TNFα levels found following sleep deprivation contributes to insulin resistance in the same way. This leads to an environment of high blood sugar and insulin insensitivity, which over time may contribute to developing higher risks of diabetes. Additionally, there might be a vicious cycle that exists in which TNFα is found at higher levels not only as a result of short sleep, but also in obese subjects even in normal sleep conditions (Figure 3). Hence, as short sleep contributes to weight gain over time, TNFα levels may climb even higher, further increasing the risk of diabetes.

Another way that short sleep might be contributing to obesity-inducing behaviors (Figure 3) is by influencing activity of different brain regions. Studies using fMRI have demonstrated that after only one night of reduced sleep, brain regions involved in reward processing showed increased activation upon exposure to images of food compared with control subjects [13,47,48]. In fact, the same reward-related brain regions that have been shown to have increased recruitment in response to food cues in obese subjects, show a similar pattern in sleep deprived subjects [49]. Additionally, insufficient sleep has been shown to reduce activity in the frontal cortex, which is associated with appetite evaluation and this may contribute to reduced impulse control when it comes to dietary choices [48]. Why might reward-processing brain regions be more prone to activation in response to food in the sleep deprived individual? Interestingly, ghrelin has been shown to activate reward processing areas in the brain upon administration to both fed and fasting subjects [50]. As discussed previously, ghrelin levels are higher in insufficient sleep conditions. Therefore, higher ghrelin levels may be contributing to the increased activation of brain reward areas in response to food, making individuals more likely to eat higher calories and spend more money on food.

Another hedonic food regulation mechanism that looks to be altered during inadequate sleep health is the endocannabinoid system. Endocannabinoids act at cannabinoid receptors in the brain and stimulate appetite; importantly, these cannabinoid receptors can be found in the reward processing brain regions [51,52]. Blocking these receptors has been shown to lead to leaner animals through decreased calorie intake [53]. Sleep restriction is associated with an increase in endocannabinoid levels, posing another avenue by which sleep insufficiency may increase calorie intake leading to obesity [53].

Substandard sleep health looks to be affecting the energy balance of calories taken in and calories burned off. The effect of insufficient sleep on energy consumed is quite well established within the literature; those who do not sleep enough have more time and opportunity to eat, disinhibition with regard to food, increased reward processing of food, etc. [54]. Additionally, there is evidence that sleep insufficiency increases the likelihood of sedentary behavior, as previously mentioned, which likely impacts energy expenditure in sleep deprived individuals. However, the impact of sleep health on the energy expenditure portion of the energy balance still requires further investigation as some conflicting results have been found [55].

Importantly, the correlation between sleep insufficiency and obesity is also bidirectional (Figure 3). People with obesity are more likely to report insomnia and trouble sleeping than those without obesity [56]. Obesity is considered a major risk factor for obstructive sleep apnea (OSA) [57,58,59]. OSA leads to feelings of excessive sleepiness and fatigue due to constant interruptions in sleep because of frequent airway collapse [58]. Individuals who are able to both lose weight and maintain their weight loss, report better sleep quality than those who lose weight but later regain it [60]. Thus, the relationship between obesity and sleep presents us with another example of a vicious cycle. An individual getting insufficient sleep is more likely to gain weight and at an increased risk of obesity due to both behavioral and physiological changes discussed previously. As they gain weight, they are increasingly likely to experience sleep disturbances like OSA, further contributing to their insufficient sleep and weight gain.

Short sleep ultimately leads to an internal environment ideal for fostering unhealthy weight gain over time by contributing to several circumstances that alter behavior. Because of the role of insulin in the production and suppression of leptin and ghrelin, respectively, the changes in these hormones during sleep deprivation are likely attributable to insulin resistance. Here, we have explained a model by which increased inflammatory markers like TNFα interfere with insulin signaling, thus altering hormone levels and promoting hunger. The inclination to increase energy intake and decrease energy expenditure creates a situation in which weight gain is expected. The more time spent in this cycle, the more likely weight gain and eventually obesity becomes. Because obesity is also a risk factor for a number of additional chronic conditions, this vicious cycle perpetuated by sleep debt is contributing to the development of a number of harmful consequences.

### 2.3. Interaction among Sleep, Nutrition, and Physical Activity

The relationship between sleep health and physical activity is interesting to consider, of course one of the first things that comes to mind is needing sufficient sleep in order to feel energized enough to engage in exercise; however, there are important processes that occur while we sleep that are prerequisites for productive physical activity. Using athletes and athletic adults as an example, we are able to see how sleep health impacts physical performance. Sleep deprivation decreased running performance, concentration of glycogen in muscles, tennis serve accuracy, sprint times, etc. [61]. Interestingly, less than 7 h of sleep has been shown to be associated with an increased risk of injury, especially when sustained for long periods of time [62]. The increase in proinflammatory cytokines that occurs during sleep deprivation is thought to impede muscle recovery and alters pain perception [61]. Additionally, disruption in circadian rhythm is associated with increased cortisol levels that could lead to a catabolic state [61].

Good sleep health is associated with improved athletic performance, therefore promoting sleep health in physically active people is likely to reduce injury and contribute to good athletic outcomes [61]. While these studies focused on athletes and active individuals, the same findings are applicable to individuals who do not currently engage in physical activity or may be looking to become more active; better sleep will help us engage in and recover from exercise. Additionally, physical activity has been proven to promote healthy sleep and may help those who struggle with their sleep to get a better night’s rest naturally [63]. Even acute exercise has been shown to be beneficial for a number of sleep health measures including total sleep time, sleep latency, and sleep efficiency [64].

Nutrition is another essential aspect of facilitating physical activity. For active individuals, maintaining specific diets that help improve their performance and fuel their workouts should be a priority. Nutrition, like sleep, is also important for ensuring recovery following training; proper nutrition helps to remake muscle glycogen stores and rebuild proteins lost during protein turnover in exercise [65]. As mentioned previously, inadequate sleep health can lead to less-than-ideal decision making when it comes to amount of food consumed and types of foods consumed. Inadequate nutrition can also contribute to feelings of fatigue, which is associated with poor physical performance [66]. The relationship between sleep and diet also looks to be bidirectional. Research has suggested that we can promote healthy sleep by consuming foods with tryptophan and melatonin [67].

Sleep health, nutrition, and physical activity are all interconnected, and inadequacy in one of these variables can lead to issues with the others. Research has already demonstrated this cyclical relationship within adolescent girls: increased sleep duration and improved sleep quality followed acute exercise, which was associated with decreased caloric intake, fat, and sugar consumption the next day [68]. It is important to work to address sleep health in order to facilitate the ability to be active and make nutritious dietary choices. On the other hand, if someone is struggling with sleep despite their efforts to improve their sleep health, diet and exercise appear to be good tools that may contribute to improving sleep. Working to address this cyclic relationship between sleep, nutrition, and exercise will undoubtedly promote better health outcomes and help to facilitate making healthy choices in each of these areas.

## 3. Sleep Chronotype and Metabolic Syndrome

Metabolic Syndrome is defined as a group of conditions that when combined increase the risks of coronary heart disease, stroke, diabetes, and other serious chronic diseases. The conditions that make up metabolic syndrome are a constellation of large waistline, high blood pressure, high blood sugar, high triglycerides, and low HDL. According to the NIH, about 1 in 3 American adults have metabolic syndrome. Many studies have demonstrated an association between the risk of metabolic syndrome and sleep duration, in which both short and long sleep duration increases the risk [69,70]. Therefore, metabolic syndrome represents another way in which sleep insufficiency is contributing to chronic disease. The mechanisms by which sleep increases the risk of metabolic syndrome can be attributed to two circumstances: either too much rapid eye movement (REM) sleep or not enough slow-wave sleep (SWS) [69]. The ratio of REM to SWS sleep is an important measure of good sleep health, and this ratio should be close to 1.

Importantly, as we discuss sleep chronotypes here, those with an evening/later chronotype describe the group of people that go to sleep later and wake up later, while a morning/earlier chronotype describe those that go to sleep earlier and wake up earlier. We are using morning and evening chronotypes to discuss the impact of sleep schedule on metabolic syndrome, because they are useful for considering how time to bed and time to wake impacts our sleep health and consequently metabolic processes. When chronotypes are given a 9 h window of opportunity to sleep between 11 p.m. and 8 a.m., they display no significant differences in time asleep or in amount of REM sleep; however, when the chronotypes are forced to sleep in a much later time window (3 a.m. to 12 p.m.), evening chronotypes get significantly more REM than they did in the earlier sleep window and compared with morning chronotypes [71]. In the later sleep window, morning chronotypes also got significantly less sleep than they did in the earlier sleep window. Taken together the results of this study demonstrate the basis of the discussions to come on sleep schedule and metabolic processes: when forced to sleep earlier, both chronotypes demonstrate similarities in relevant sleep measures including amount of REM sleep, when forced to sleep later, differences arise. Those more used to later sleeping schedules get more REM and more total sleep time while those who tend to sleep earlier get less total sleep and less REM. It is the time to sleep that is relevant here in our discussion on metabolic syndrome, and evening chronotypes represent the group of people who tend to go to sleep later and wake up later.

During REM sleep, our heart rates, respiration rates, and blood pressure all increase irregularly and it is in this sleep stage that we dream [72]. REM sleep is often referred to as paradoxical sleep because it resembles unconscious wakefulness. Importantly, as the night progresses, we spend more time in REM sleep. Harfmann et al. demonstrated that those with later chronotypes spent more time in REM sleep [73]. This becomes troublesome for individuals who prefer to sleep late because it is known that awaking during a REM stage can make a person feel more tired [74]. Therefore, the later one sleeps, the higher the chance that he or she will be aroused during a REM sleep stage and feel groggier during their day. Additionally, this will increase the likelihood of sedentary behavior. Research has demonstrated that individuals with an evening chronotype, meaning they go to sleep later and wake up later, demonstrated lower physical activity and higher sedentary time [75]. In fact, those with an evening chronotype are also at a higher risk of cardiovascular diseases (CVD), type 2 diabetes, and obesity (Figure 4) [75,76].

Because individuals with a later chronotype spend more time in REM, they spend more time during their night with a higher heart rate and blood pressure. Furthermore, cortisol levels peak in the morning in order to help us wake up and feel energized to begin our day. To do this, cortisol promotes the stress response of increased blood sugar and an increase in the drive of the sympathetic nervous system [77,78]. When one sleeps past that peak, this cortisol is not used as effectively as it would be if one was awake and active. Therefore, the lingering cortisol in late risers instead will create these effects within the sleeping individual, driving their heart rate, temperature, and blood pressure higher than it should be for a restorative sleep. If someone is a habitual late riser, this will be a prolonged exposure to the stress response during sleep (Figure 4). Additionally, it has been shown that it is during early morning sleep that acute exacerbations of a number of chronic disorders occur most frequently [69]. Smiley et al. reviewed how the highest incidence of myocardial infarction is during early morning sleep; Additionally, flare ups in IBS, hypertension, etc. are also not uncommon during early morning sleep. Importantly, one study demonstrated that early morning light administration improved depression when compared with a control group. Together, this demonstrates that it is possible that many of these early morning events may have been preventable had these patients been habitual early risers [69]. 

SWS, as opposed to REM sleep, occurs in longer stages earlier on in the night. Those who go to sleep earlier will therefore spend more time in SWS. In further opposition with REM, SWS is characterized by a very low heart rate, respiratory rate, and blood pressure. A number of essential biochemical processes take place during SWS. For example, growth hormone (GH) is released during SWS [79]. GH facilitates fat burning, bone building, repair, regeneration, and promotes lean body mass.

Growth hormone releasing hormone (GHRH) also promotes the activity of astrocytes in the brain, which means that during SWS sleep, astrocytes are more active in cleaning up β amyloid and tau proteins [80,81,82]. Those with an evening chronotype thus will have suboptimal levels of clearance of β amyloid and tau protein due to the decreased duration of SWS. Research has shown that sleep deprivation is associated with higher levels of β amyloid [83]. Importantly, β amyloid and tau protein are both implicated in the neuropathology of Alzheimer’s Disease (AD), which is the number one cause of dementia. One night of sleep deprivation leads to impaired functioning of the hippocampus, which plays an essential role in forming and storing memories [84]. Therefore, SWS is important for both memory processing and learning [85]. Unsurprisingly, research has shown a positive correlation between evening chronotype (those who spend less time in SWS) and dementia [86]. Taken together, these studies show that sleep is implicated in memory, consolidation, and long-term cognitive ability (Figure 4).

Additionally, ghrelin release peaks at the beginning of the night when SWS occurs. An individual with an evening chronotype will end up suppressing this important release of ghrelin by staying awake later. Ghrelin has been shown to have a protective effect against endothelial damage caused by lipoproteins known as advanced glycation end products (AGEs) [87]. AGE levels are also higher in chronic sleep insufficiency conditions [88]. The presence of AGEs is a result of lipoproteins being exposed to sugars commonly seen in atherosclerosis and diabetes. Thus, it is possible that as sleep insufficiency contributes to the development of CVD and diabetes, it will also create conditions ideal for AGE inflicted endothelial damage in later stages (Figure 4).

Though we have emphasized the important consequences of decreased SWS and increased REM in association with metabolic syndrome here, it is important to point out a potential role for insufficient REM as well. Importantly, insufficient REM is associated with elevated BMI and being overweight [89]. While BMI is not a specific component of metabolic syndrome, being overweight and/or obese as measured by BMI likely indicates a large waist circumference, higher blood pressures, and higher fasting blood sugar which are all parts of the criteria for metabolic syndrome. Another consequence of insufficient REM is an increased risk of dementia (similar to insufficient SWS), which indicates that processes that occur during REM are also important for memory [90]. The mechanisms behind this association are still not clear, though, and should be investigated further.

The duration of time we spend asleep is not the only piece of the puzzle; what time we go to bed and what time we wake up also contribute to our overall sleep health. As discussed, it is best to go to sleep earlier and wake up earlier. According to Dr. Matt Walker of the Sleep and Neuroimaging Lab at the University of California Berkeley, Berkeley, CA, USA, “every hour of sleep before midnight, is worth two after midnight” [91].

### 3.1. Sleep Fragmentation and Metabolic Syndrome

Sleep fragmentation, defined as repetitive short interruptions of sleep, is another avenue by which individuals may experience disturbances in their sleep architecture that alters the amount of time spent in necessary sleep stages. For example, REM sleep fragmentation describes the experience of arousals that disturb the REM sleep stage specifically, thus leading to insufficient REM. It has been shown that higher levels of sleep disturbances is associated with increased risk of hypertension and non-HDL lipoprotein cholesterol in adolescents [92]. Additionally, greater sleep fragmentation is significantly associated with higher BMI and decreased insulin sensitivity [93,94]. It is thought that the effects of disrupted sleep on insulin sensitivity and glucose metabolism is mediated by the effects of sleep fragmentation on the sympathetic nervous system. It is well established that sleep fragmentation leads to increased autonomic sympathetic activation [94,95]. Medic et al. explains that the long-term risks of sleep disruption include cardiovascular disease, metabolic syndrome, type II diabetes mellitus, dyslipidemia, etc. [95].

### 3.2. Circadian Misalignment and Metabolic Syndrome

Circadian misalignment is another interesting piece of this puzzle worth discussing. Circadian misalignment is a term that refers to disruption of normal circadian rhythm, which are 24 h cycles that dictate our physiology and behaviors independent of our external environment or cues. Circadian misalignment can include inappropriate timing of sleeping, eating, and other behaviors. Shift work is one example of a driving force behind circadian misalignment: as shifts rotate and workers are forced to constantly alter their working hours and free time, their internal cues are thrown off [96]. It has been demonstrated that circadian misalignment in younger subjects is a risk factor for lifelong obesity and that it affects the energy balance, increasing the chance of weight gain [97]. Circadian misalignment has been shown to be associated with some of the components of metabolic syndrome, including increased blood pressure as well as higher postprandial glucose and insulin [98,99]. Additionally, Karlsson et al. revealed that among shift workers, high triglycerides, low HDL, and obesity clustered together more often than in day workers [100]. Interestingly, evening chronotype appears to be more at risk of circadian misalignment. Social jet lag, a term used to describe the discrepancy between sleep schedule on the weekdays versus on the weekends, has been found to be significantly associated with eveningness (or later chronotype) as evaluated by the Morningness-Eveningness Questionnaire [101]. This further demonstrates the risks presented by having a late sleep schedule, and risking creating disruptions of circadian rhythm.

### 3.3. Sleep Duration and Metabolic Syndrome

Sleep duration has been shown to be related to both metabolic syndrome and metabolic syndrome severity in a U-shaped association [102]. Here, Smiley et al., used the generalized additive model to demonstrate that those who slept 7 h had the lowest risk of metabolic syndrome, while those who slept both less than and more than 7 h had increased risks. A similar pattern was seen for metabolic syndrome severity score: those sleeping 7 h showed the lowest severity while those sleeping either less than 5 h or more than 9 h showed nearly similar means of metabolic syndrome severity. The data supporting this relationship is robust, as this U-shaped association has been found using three different national datasets to study it: Jackson Heart Study (JHS), Reasons for Geographic and Racial Differences in Stroke (REGARDS), and National Health and Nutrition Examination Survey 2013/2014. All three datasets were evaluated individually and showed similar U-shaped association between sleep duration and the risk of metabolic syndrome and its severity [102,103].

Importantly, research has shown that sleep duration has a U-shaped association with many of the components of metabolic syndrome including waist circumference, fasting blood sugar, systolic BP, diastolic BP, and triglycerides [103,104]. Sleep duration also demonstrated an inverse U shaped association with HDL [103,104]. All of these are important components of metabolic syndrome. Through the effects of sleep duration on the components of metabolic syndrome, sleep duration also looks to contribute to the development of and severity of metabolic syndrome. More research is warranted to further clarify the mechanisms behind each component and their relation both with each other and with sleep duration.

## 4. Sleep Health and Immunity: Inflammation

The existence of a relationship between insufficient sleep and inflammation has been well documented in existing literature. This is important because inflammation is the basis of many diseases, as we will see in the coming sections. Sleep loss has been shown to lead to higher levels of inflammatory signaling, specifically through cytokines like TNFα, IL-6, and C reactive protein (CRP) [11,105,106]. It is thought that IL-6 might play a role in sleep regulation, based on a study that found that sleep deprivation changed the pattern of IL-6 secretion causing daytime oversecretion and nighttime undersecretion [107]. Vgontzas et al., hypothesized that this finding may mean that the high daytime IL-6 levels could be responsible for increased daytime fatigue in sleep deprived subjects, but their low nighttime IL-6 levels may have helped contribute to their deeper sleep the following night [107]. The effects of sleep insufficiency on IL-6, may also contribute to elevated CRP seen following sleep loss because IL-6 can induce CRP synthesis [107,108]. Therefore, someone with chronic sleep insufficiency, may be exposed to high IL-6 levels for prolonged periods of time, which may be responsible for increasing levels of CRP. Additionally, sleep fragmentation is associated with inflammatory signaling molecules including CRP [109].

It is hypothesized that one of the ways in which insufficient sleep or oversleeping contributes to inflammation is through vascular dysfunction. Increased blood pressures are seen following sleep loss, and it is known that high blood pressures can lead to endothelial shear stress [110]. Chae et al. demonstrated that as blood pressure climbs so does IL-6, along with other inflammatory markers [111]. Therefore, increased autonomic activation may be one of the main mechanisms by which sleep insufficiency is contributing to inflammation (Figure 5).

## 5. Contributions of Inadequate Sleep Health on Other Chronic Diseases

Obesity and depression/anxiety/stress are only a few of the conditions that appear to have a worrisome relationship with sleep insufficiency. As time asleep has fallen, many other chronic conditions have increased as we saw with obesity and mental distress. As we will see, together, obesity and depression are both in themselves mechanisms by which short sleep is contributing to the prevalence of other chronic diseases.

### 5.1. Sleep Health and Depression/Anxiety/Stress

The relationship that exists between sleep insufficiency and mental illness including depression and anxiety is well documented. According to Blackwelder et al., inadequate sleep is associated with significant odds of frequent mental distress [7]. This relationship has also been demonstrated between sleep quality and increased stress levels [112]. Insomnia, a sleep disorder in which individuals have a difficult time falling and/or staying asleep, is known to be associated with increased risk of depression [6,113]. Additionally, evidence shows that as the severity of insomnia increases, the risk of depression increases as well [114]. Insomnia is a good model for studying the effects of sleep deprivation because even though an insomniac may spend a normal amount of time in bed, their sleep quality and duration is worse. If sleep insufficiency influences the risk of mental distress, what are the mechanisms by which these risks are increased?

The effects of short sleep on cortisol levels are an interesting consideration for how lack of sleep might contribute to mental distress. Cortisol, the fight or flight hormone, is known to be a biomarker for stress levels; high cortisol levels can indicate high stress [115]. Importantly, cortisol is also inextricably linked with the sleep cycle. Cortisol begins to rise in the middle of the night while we sleep, peaking right as we wake up in order to help us become alert for the new day [116]. Sleep loss has been shown to increase cortisol levels [117]. Importantly, high cortisol is known to be a risk factor for depression and some with major depressive disorder (MDD) have significantly higher cortisol levels [118,119,120]. Studies on the cause of high cortisol in subjects with depression have found that despite significantly higher cortisol levels, there was no significant difference in ACTH levels [121]. This finding indicates that misregulation of the HPA axis is not occurring at the level of the hypothalamus, but instead might be occurring at the level of the pituitary or adrenal glands. In MDD, abnormal functioning of the glucocorticoid receptor (GR) has been demonstrated, which may contribute to hypersecretion of cortisol [122].

It is possible that sleep loss creates a high cortisol environment like the one seen in patients with high stress levels and/or depression. As mentioned previously, sleep loss is known to have an inflammatory effect by increasing the levels of some inflammatory cytokines and signaling molecules [40,105]. For example, cell nuclear factor (NF)-κB activation was significantly higher following sleep loss compared with a full sleep night [105]. This is significant because NF-κB has been shown to interfere with GR activity by disrupting its translocation to the cell membrane as well as its phosphorylation status [123]. NF-κB is not the only signaling molecule affected by short sleep duration that may impact cortisol secretion. As discussed previously, TNFα is also found at increased levels in sleep deprivation conditions [40,41]. Research demonstrates that TNFα can interfere with GR-induced gene expression and is thus associated with glucocorticoid resistance [124].

The effects of these cytokines on GR functioning are important because it might explain how sleep is contributing to climbing cortisol levels (Figure 6). As these cytokines reach higher concentrations following sleep loss, GR might become less and less responsive to cortisol as explained above. Cortisol at high levels should normally decrease HPA axis activity in a negative feedback system, but due to GR dysfunction this negative feedback system will be lost, and cortisol levels will continue to climb. As cortisol levels accumulate, feelings of stress will climb and the risk of mental distress like depression climbs with it.

As with obesity, the relationship between sleep and mental distress is also bidirectional. Psychosocial stress has been shown to lead to consistent increases in sleep disturbances [125]. For patients suffering from depression, quality and duration of sleep is a common struggle. In fact, a number of studies reports that more than half of their subject groups of depressed patients experience sleep disturbances in some form [126,127,128]. Additionally, the duration of insomnia is positively correlated with both the duration of the depressive episode and the number of depressive episodes [126,129]. Presence of severe insomnia has also been shown to be associated with a more severe presentation of MDD [129].

Taken together, this information with the previously discussed relationship between short sleep and the increased risk of mental distress, a vicious cycle in which one feeds into the other is elucidated. For an individual who struggles to get sufficient amounts and quality of sleep, the risk of developing depression/anxiety/stress is increased. If this individual does develop one or more of these conditions, their likelihood of continuing to struggle with getting a good night’s sleep will be increased. In fact, their ability to sleep may even worsen. This then may cause their mental health episode to present with more severe characteristics, last longer, and may increase the likelihood of recurrence. Therefore, addressing issues with sleep may not only prevent development of a mental illness, but may also decrease severity of symptoms and promote quicker improvement.

### 5.2. Diabetes Mellitus

Diabetes is another example of a chronic disease that is on the rise; between 1995–1997 and 2005–2007 alone, the age-adjusted incidence of diabetes in the United States increased by 90% [130]. In 1958, the percentage of the US population with diagnosed diabetes was less than 1%; in 2019, that number had reached 11.3% [131,132]. In the United States, diabetes ranks as the 8th most common cause of death, and it is the most expensive chronic condition with the estimated annual cost of diabetes reaching $327 billion according to the CDC.

Research demonstrates that as duration of sleep decreases, the risk of diabetes simultaneously increases [11,133]. Many of the mechanisms explained previously likely play a role in the relationship between sleep and diabetes as well. For example, the multiple ways in which sleep loss contributes to obesity, also account for the relationship between sleep and diabetes. This is because obesity is one of the most important risk factors for diabetes; almost half of adult diabetics in the United States are obese [134]. More specifically, the increase in cytokine release seen following short sleep or sleep disturbances may be creating chronic, unresolved immune challenges [40,135,136]. Importantly, as previously mentioned, this is likely contributing to the reduction in insulin sensitivity seen in sleep-deprived subjects, a hallmark of diabetes (Figure 3). Additionally, a significant relationship between sleep deprivation and increased hemoglobin A1C (HbA1C) does exist [137,138,139].

With the rapidly growing prevalence of diabetes in the United States, it is essential to consider modifiable risk factors of the disease and to encourage individuals to make lifestyle changes early to prevent disease development. Considered all together, evidence supports the model that sleep deprivation contributes to metabolic changes that both increase weight and simultaneously drive up an individual’s risk of diabetes (Figure 3). Therefore, individuals sacrificing their sleep for other activities should consider how they might be impacting their long-term health.

### 5.3. Hypertension

According to the CDC, 47% of adults in the United States had high blood pressure as of 2017. Between 1988 and 1991, 24% of the adult population in the United States had hypertension according to the National Health and Nutrition Examination Survey [140]. In only 30 years, the incidence of hypertension has nearly doubled and today almost one out of every two people has high blood pressure, which is a staggering increase. A relationship in which short sleep duration increases the risks of hypertension has been established by a number of studies [11,141,142]. Interestingly, this relationship was shown to be U-shaped, where long sleep duration also was associated with increased risks of hypertension.

It is thought that the effects of sleep insufficiency on sympathetic nervous system (SNS) activity might be one mechanism by which sleep deprivation contributes to hypertension. One study demonstrated that both blood pressure and heart rate were significantly higher on a sleep-insufficient day compared with a normal day [143]. Another study found that while blood pressure was significantly increased following sleep deprivation, heart rate was not [144]. The picture of how sleep deprivation contributes to hypertension is not as clear as a simple increase in sympathetic control of the cardiovascular system.

Sleep deprivation is also associated with an increased production of endothelin, which is a vasoconstrictor [145,146]. It is possible that long-term, habitual sleep insufficiency may contribute to an accumulation of endothelin which consequently constricts vessels and increases blood pressure. While sleep loss appears to be increasing this potent vasoconstrictor, it also might be decreasing an important vasodilator. Evidence suggests that total sleep deprivation is associated with a reduction in the activity of nitric oxide synthase in rats [147]. Nitric oxide is important for inhibiting sympathetic tone and promoting vasodilation. Therefore, sleep deprivation may be increasing blood pressure through a two-pronged approach (Figure 7).

### 5.4. Other Cardiovascular Diseases

Cardiovascular diseases (CVD) include a group of conditions that affect the heart and blood vessels. The diseases that fall into this category include coronary heart disease, cerebrovascular disease, peripheral artery disease, and rheumatic heart disease to name a few. Heart disease is the leading cause of death in the United States, with one person dying every 36 s [148]. According to data from the 2017 National Health Interview Survey, the prevalence of all kinds of heart disease was approximately 10.6%. Importantly, the CDC reports that heart disease is the leading cause of death in the United States.

According to some studies, both short and long sleep duration are associated with modestly increased risks of coronary heart disease [11,149]. However, Lao et al., found coronary heart disease has been shown to be significantly associated with short sleep duration and poor sleep quality [150]. Ferrie et al. demonstrated that decreased or increased sleep duration was associated with increased risk of mortality [151].

One of the ways inadequate sleep health may be contributing to CVD is through its effect on inflammatory proteins. For example, sleep deprivation has been shown to lead to an increase in the levels of interleukin (IL)-6 [106]. This is important because IL-6 is known to contribute to C reactive protein (CRP) synthesis in the liver, and CRP levels have additionally been shown to rise following sleep deprivation [108,152]. High CRP is associated with increased risks of coronary heart disease [153]. Additionally, the impact of insufficient sleep on these inflammatory markers may be contributing to the development of atherosclerosis, which is associated with inflammation of blood vessel walls [154]. Inflammation can be associated with all stages in the development and progression of atherosclerosis; in fact, elevated values of certain inflammatory markers, including IL-6 and CRP have been associated with adverse prognosis in patients with atherosclerosis [155]. Importantly, atherosclerosis is considered a main cause of CVD [156]. Thus, inadequate sleep health looks to be increasing the risks of CVD by potentially increasing the odds of developing atherosclerosis because of its inflammatory effects (Figure 8).

In addition to a direct effect on CVD development, sleep insufficiency is likely also contributing to CVD indirectly through its previously discussed influence on other chronic conditions (Figure 8). For example, both hypertension and obesity have been shown to be major risk factors for CVD [157]. Therefore, by increasing the risk of these separate chronic conditions, substandard sleep health simultaneously increases the risk of CVD.

### 5.5. Cancers

Poor sleep quality has been shown to be associated with increased long-term risks of cancer development in a study that looked at more than 10,000 individuals aged 50 and older [158]. Another study, which involved more than half a million subjects, found that in individuals who suffer from insomnia, there was a 24% increased risk of cancer [159]. The relationship between sleep health and cancer is also bidirectional; cancer patients commonly experience poor sleep due to disturbances like insomnia, nightmares, etc. [160].

It is thought that melatonin may have a protective role against cancer. Evidence demonstrates that it arrests cancer at the initiation, progression, and metastasis phases [161]. Therefore, one potential mechanism by which substandard sleep health contributes to cancer development may be lower levels of melatonin. Those with an evening chronotype or those who do shift work, for example, may be more exposed to light at night when they should be in the dark, and produce less melatonin as a result [162]. Another potential mechanism may be attributable to vitamin D. It is possible that those who sleep during the day, for example those with evening chronotypes or who work in shift work, are exposed to less natural light, and therefore might produce lower levels of vitamin D [162]. There is some evidence for a protective role of vitamin D against some cancers, like breast cancer [163]. The third path between sleep insufficiency and cancer is through long-term inflammation [164]. The combined effects of potentially chronic low levels of melatonin and vitamin D as well as long term inflammation may play a role in increasing the risk of cancer in sleep deprived individuals (Figure 9).

Another potential mechanism by which sleep insufficiency manifests as a risk factor for cancer is through its effect on immune functioning. According to Liu et al., sleep loss can lead to dysfunction in immune surveillance for cancer cells; insufficient sleep was associated with increased levels of profillin 1 (PFN1), a negative regulator of cytotoxic immune cell functioning [165]. To exacerbate this further, sleep deprivation has been shown to increase NF-κB, which is reported at higher levels in tumor samples and can promote oncogenic processes [105,166]. This combined environment of increased levels of a pro-oncogenic transcription factor (NF-κB) with diminished immune surveillance creates a high-risk environment for cancer development (Figure 9).

### 5.6. Pulmonary Diseases

Obstructive sleep apnea (OSA), a disease we have already mentioned in this paper, and chronic obstructive pulmonary disease (COPD) are both pulmonary diseases that are interconnected with sleep. OSA is characterized by frequent airway collapses during sleep and is therefore in itself a sleep disturbance [167]. Sleep also may contribute to the development of OSA by increasing the risk of obesity, a primary risk factor for OSA [167]. When it comes to COPD, sleep quality is often overlooked as a symptom, however it is estimated that the prevalence of sleep disturbances may be more than 75% in COPD patients [168]. Sleep may also be connected to idiopathic pulmonary fibrosis (IPF) through its effects on inflammation, because inflammation could contribute to fibrosis through its promotion of airway remodeling leading to abnormal collagen deposition (Figure 10) [169,170]. Interestingly, IL-6 is seen at higher levels in COPD patients and has been implicated in the pathogenesis of IPF, and as we have discussed also rises following sleep deprivation [171,172].

Pneumonia, another pulmonary disease, is an infection within the lungs’ alveoli. Patel et al. has demonstrated that both short and long habitual sleep are associated with higher risks of pneumonia [173]. It has been established that sleep deprivation can make individuals more susceptible to infection by weakening the immune system. Both natural killer cell and T cell immune responses were diminished following insufficient sleep [174,175]. This may be because sleep deprivation has been shown to increase PFN1, a key protein involved in the downregulation of cytotoxic cells, as mentioned previously [165]. Therefore, substandard sleep health may put individuals at a higher risk of pneumonia by making them more susceptible to infections (Figure 10).

### 5.7. Gastrointestinal Diseases

Sleep insufficiency is thought to influence gastrointestinal diseases through its impact on inflammation. As previously discussed, poor sleep can lead to an increase in inflammatory markers like TNFα and IL-6 [106]. Importantly, these cytokines have been implicated in inflammatory bowel disease (IBD) and irritable bowel syndrome (IBS) [176]. Kim et al. found that shift workers demonstrated significantly higher levels of IBS [177]. Therefore, with this evidence that substandard sleep health is associated with higher levels of this gastrointestinal condition we can surmise that it may be through sleep’s effect on inflammatory responses that it contributes to diseases like IBS and IBD (Figure 11).

Poor sleep quality has also been shown to affect other gastrointestinal diseases. For example, some studies have shown that sleep insufficiency can worsen the symptoms of gastroesophageal reflux disease (GERD), by exacerbating and leading to longer durations of reflux [178]. Additionally, poor sleep leads to hyperalgesia in patients suffering from GERD [179]. Interestingly, it has been shown that increased sensitivity to pain can be related to feelings of stress and anxiety. For example, Dickhaus et al. demonstrated that IBS patients gave higher ratings of unpleasantness in response to visceral stimuli while exposed to auditory stress conditions [180]. As previously discussed, there is a well-established relationship between sleep insufficiency and stress/anxiety. It is possible that one of the ways sleep insufficiency contributes to hyperalgesia in gastrointestinal diseases is by increasing stress levels (Figure 11). Therefore, we have evidence that sleep insufficiency may not only contribute to the development of gastrointestinal diseases but can also worsen their manifestations.

### 5.8. Neurological Diseases

Quality sleep is important for maintaining neural health. As discussed previously, it is during SWS that misfolded neurotoxin proteins are cleared from neural tissue [80,81]. This is because GHRH released during SWS helps promote astrocyte activity in the brain [79]. Unsurprisingly, research has deduced a bidirectional relationship between insufficient sleep and Alzheimer’s disease (AD) [181]. Importantly, Brown et al., demonstrated that the relationship between longer sleep latency and increased levels of β amyloid was independent of the APOε4 allele, an allele known to increase the risk of AD [182].

Interestingly, GH has also been implicated in amyotrophic lateral sclerosis (ALS); GH insufficiency has been noted in ALS and GH has proven to have therapeutic effects in ALS mouse models [183]. GH is considered to be neuroprotective against ALS, but the promising findings seen in animal models have thus far failed to translate in clinical trials. Regardless, the relationship between GH and ALS should still be considered for the purposes of this paper, where we have discussed how SWS is important for GH production and the overall relationship between sleep and chronic disease.

Sleep disturbances are additionally highly linked with Parkinson’s Disease (PD). It has long been known that PD patients struggle to get good sleep, in fact sleep disorders are one of the most common non-motor manifestations of PD [184]. Some of the more common manifestations of sleep disorders in patients with PD include REM sleep behavior disorder, restless leg syndrome, and sleep apnea and all together these disturbances contribute to many PD patients experiencing excessive daytime sleepiness [185]. New research is finding that sleep insufficiency is not only a manifestation of PD, but also appears to be a risk factor. Hsiao et al. looked at more than 90,000 subjects and found that non-apnea sleep disorders are associated with increased risks of PD [186]. Like AD, PD is also known to be related to a buildup of neurotoxic protein; it is thought that oligomers of α-synuclein disrupt cell functioning and lead to neuronal cell death [187]. With this in mind, it is possible that sleep insufficiency contributes to PD in a similar mechanism as it does to AD, where the unfolded protein response (UPR) that would normally help to clear misfolded proteins requires enough time in restorative SWS to efficiently do its job (Figure 12).

### 5.9. Renal Diseases

While research in this field is still limited, there are a number of studies that have looked at the relationship between sleep health and renal diseases. For example, Bo et al., found that those with a poor sleep profile experienced an increased risk of developing chronic kidney disease (CKD). They also saw that subjects with either short or prolonged sleep duration both had increased risks of CKD [188]. There are a few mechanisms by which sleep insufficiency may be contributing to CKD development that warrant further investigation. As discussed previously, sleep insufficiency is known to have an activating effect on the SNS [143]. Interestingly, it is well-established that the SNS is a contributor to CKD development and progression [189]. Another study has demonstrated that short sleep, especially less than 5 h, is a positive predictor of proteinuria (increased levels of protein in the urine) [190]. Studies have demonstrated that increased blood pressure is associated with increased risk of persistent proteinuria [191]. Proteinuria also happens to be a strong indicator of CKD as it indicates glomerular disease or dysfunction of renal tubules [192]. Therefore, it appears that by increasing blood pressure, sleep insufficiency increases the risks of CKD by creating a higher risk of persistent proteinuria (Figure 13). Sleep insufficiency also likely contributes to CKD indirectly by increasing the risks of diabetes, high blood pressure, and obesity, which are all independent risk factors for CKD.

### 5.10. Musculoskeletal Diseases

Patients who suffer from osteoarthritis commonly experience sleep difficulties, largely as a result of pain. Studies have revealed that the relationship between pain and sleep in patients with osteoarthritis is bidirectional [193]. Pain keeps individuals awake at night and then lack of sleep makes their pain worse, creating a vicious cycle (Figure 14). Technically, osteoarthritis is classified as a non-inflammatory arthritis. However, there is still inflammatory elements to osteoarthritis like chronic synovitis. Inflammation is associated with increased pain [194]. It is possible that inflammation is a mechanism by which sleep insufficiency makes this condition less tolerable. It is known that IL-6 is seen at higher levels in osteoarthritis [195]. We previously explained that IL-6 is also increased following sleep insufficiency, and thus may represent a possible mechanism by which sleep affects the pathogenesis of osteoarthritis (Figure 14) [106].

Interestingly, IL-6 also looks to be implicated in both rheumatoid arthritis (RA) and dermatomyositis, which is an inflammatory muscle disease that involves rashes to the skin [196]. Patients with RA tend to report high levels of insufficient sleep, often as a result of pain levels [197]. For RA patients, their level of functional disability was associated with their quality of sleep; further, higher levels of disability were more likely to lead to depression, which could further disturb sleep [197]. Therefore, lack of quality sleep and its relationship with pain seems to worsen the clinical presentation of RA (Figure 14).

### 5.11. An Additional Consideration

Another point of emphasis is that the detrimental effects of inadequate sleep health covered here do not require decades of habitual short sleep to manifest. In fact, one study focusing on shift work and its disruptions of one’s sleep cycle found that it only took on average 3.97 years of shift working to see the negative physiological effects [40]. Some of the mechanisms of short sleep discussed above take only one night of sleep deprivation. Thus, it is important to avoid sleep disturbances and work to improve sleep schedules early in one’s life to avoid the effects of chronic sleep debt (Figure 15).

### 5.12. Sleep Schedule Variations

Our sleep schedule is largely influenced by the light and dark cycles of the sun in our environment. The suprachiasmatic nucleus within the hypothalamus responds to the presence or absence of light, and signals to the pineal gland to produce melatonin when the time comes to sleep [198]. The amount of sunlight we get each day is dependent on our location and time of year. In the summer months we are afforded more hours of daylight, while in the winter months we spend more time in the dark. As a result, our bodies need less sleep in the summertime than they do in the wintertime. Research has demonstrated that during summer, individuals get less sleep than they do in the winter [199]. The high arctic offers itself as a unique case study for how light impacts the circadian rhythm and sleep of its human residents. In the arctic region, the summer months lack the darkness that would typically signal us to secrete melatonin to promote sleep. In the winter months, on the other hand, there is no daylight. Thus, it is unsurprising that residents of this region suffer from poor sleep during the winter and summer months [200].

Because our sleep schedules are so dependent on season and location, the schedule that is optimal for someone living in New York City in June will be quite different from the schedule that is optimal for someone living in Argentina in June. This is because in the northern hemisphere, where New York City is found, June marks the beginning of the summer season, meaning many hours of daylight and less time needed to sleep. In the southern hemisphere, where Argentina is found, on the other hand, June marks the beginning of winter, so Argentinians will see less daylight and need to spend more time sleeping. Therefore, it is important to note that while there are guidelines recommending how much sleep adults should aim for, a healthy sleep schedule will also be impacted by other factors, and it is important to listen to our bodies to find what is optimal for our individual selves.

Importantly, despite all of the evidence for the risks associated with insufficient sleep, it still must be noted that sleep is not a one size fits all entity. For example, there is a small subset of individuals that naturally sleep for less than 6.5 h at night and do not experience daytime sleepiness or the cognitive symptoms that are associated with disease progression [201]. This subset is a group referred to as Familial Natural Short Sleepers and there are a number of mutations associated with this phenomenon [202,203]. There is additional evidence that there is a subset of people who naturally require longer hours of sleep. These habitual long sleepers do not seem to differ with respect to homeostatic sleep regulation, and studies suggest these differences in sleep duration are attributable to the circadian pacemaker (found within the suprachiasmatic nucleus of the hypothalamus) [204,205]. Natural short sleepers and long sleepers are the exception and not the rule. The rule is explained by a bell-shaped epidemiological pattern. As shown in Figure 16, mean ± two standard deviation covers 95% of the population. Therefore, only the 2.5% below and the 2.5% above the 95% are considered the exceptions. Considering 6–8 h of sleep as the 95% confidence interval for sleep duration, natural short sleepers are among the 2.5% of population that are below the 95%. Similarly, natural long sleepers are among the 2.5% of population that are above the 95%. These exceptions should not be the reference of lifestyle for the general population. Therefore, it is important to share the findings of evidence-based medicine which is backed up by solid science and is applicable to almost everybody.

## 6. Benefits of Quality Sleep on Chronic Diseases

Here, we have reviewed the detrimental effects of insufficient sleep on our health, specifically through its contributions to chronic diseases. Now we will address the benefits of quality sleep not only on our objective health, but also on our quality of life.

While it is clear by now that insufficient sleep can lead to inflammation, it is important to mention that quality sleep is key for proper functioning of our immune systems [206]. Lange et al., demonstrated that quality sleep may improve adaptive cellular immune responses by helping to balance the oscillation between type 1 and type 2 cytokine activity [207]. Type 1 cytokines are pro-inflammatory while type 2 cytokines are anti-inflammatory and striking a balance between these two categories of cytokines is important to prevent inflammation that could cause tissue damage; a healthy sleep schedule appears to play a role in maintaining this balance. Additionally, studies in mice models have shown that sleep strengthens the innate immune system in both numbers of cells and in function in defense against bacterial pathogens [208]. Therefore, improving sleep is a good way to help fight damaging inflammation and strengthen immunity.

Improving sleep quality has been shown to improve mental health, including a reduction in depression, anxiety, and stress [209]. In fact, targeting symptoms of insomnia, both pharmacologically and non-pharmacologically, can affect the trajectory of depressive episodes by reducing symptoms and duration [210].

Adequate sleep is also a tool that can be employed in weight management. Better sleep quality increases the likelihood of successful weight loss in subjects trying to lose weight [211,212]. Research has demonstrated that when overweight subjects who are accustomed to habitual short sleep undergo a period of sleep extension, they reduce their caloric intake [213]. While research in this area is still in the early stages, studies to date seem to demonstrate that promoting sleep extension in habitual short sleepers may improve cardiometabolic risk [214]. Therefore, focusing on improving quality and duration of sleep may be a good tool for weight loss and maintenance.

## 7. Relationship between Sleep Health and Wellness

Short sleep duration has been shown to be associated with lower levels of happiness [215]. Importantly, Zhao et al. demonstrated that these findings were in line with existing literature that demonstrates an association of inadequate sleep health with mental health and wellbeing, but in addition found that the relationship with happiness was independent of mental health [215]. They hypothesize that this might indicate that other consequences of substandard sleep health such as physical discomfort and fatigue may affect happiness. Higher self-perceptions of well-being, a term that here includes feeling good and functioning well, are associated with better health outcomes [216,217].

Overall, a quality sleep schedule leads to a better self-perception of wellbeing. An important aspect of wellbeing that often goes overlooked is connection to nature, and this variable is also connected with sleep health. In a study that looked at trail users versus non-trail users, it was found that those who did engage with nature on trails experienced less sleep troubles than those who did not [218]. Although trail use correlated with better sleep health, sleep health was significantly associated with better self-rated wellness and health, independent of using or not using the trail [219]. This means connecting to nature and improving sleep health may have synergistic effects on self-perceived wellbeing. Additionally, encouraging individuals to spend more time engaging in physical activity outdoors may be a good tool to help improve both sleep health and self-perceptions of wellbeing.

Better sleep quality is also significantly correlated with higher self-ratings of mood [220]. Additionally, people who sleep well were found to be more satisfied with life as opposed to those who struggle with sleep, who were more likely to view happiness with a zero-sum mindset [221]. Thus, emphasizing sleep quality may present as a useful tool in helping individuals maintain a positive outlook on life.

## 8. Measuring Sleep Quality

When a patient comes to their physician with a problem, physicians try to measure the characteristics of their symptoms and the clinical presentation of the patient. For example, asking when the patient has pain, exploring the provoking factors, quality of pain, radiation of pain, severity of pain on a scale of 1 to 10, and time duration of pain (PQRST mnemonic) is essential. How can physicians assess sleep insufficiency? Several questionnaires have been developed to assess the quality of sleep. The mini-sleep questionnaire (MSQ) is one of the most practical questionnaires that can be easily used in everyday clinical practice [222]. The MSQ allows for evaluation of both sleep quality and daytime sleepiness through only a brief (10 item) survey [223]. Within research, the MSQ has been used to evaluate how sleep may be associated with self-perception of wellness and health [224,225]. For example, Tabaraii et al. used the MSQ to demonstrate that in patients with rheumatoid arthritis, long sleep duration, long naps, and decreased sleep quality are associated with lower self-ratings of wellness and health [226].

Using the MSQ can help physicians pinpoint the underlying cause of inadequate sleep health, as they would attempt to do with any other complaint a patient may present with. This will facilitate the ability of physicians to assist their patients in addressing the issues that are unique to their sleep health. This will be a more effective way to encourage patients to address their sleep health because it demonstrates to patients that their physician is taking the time to learn what their unique circumstances are, which will help create a rapport in which the patient and physician can work together to improve their sleep health. Furthermore, instead of physicians having to provide patients with a complete list of lifestyle modifications to improve their sleep, using the MSQ they can instead focus narrowly on the specific issue harming their patients’ sleep.

## 9. Sleep Education

Considering the demonstrated benefits of a quality sleep schedule, it is important to discuss how effective sleep education programs are. While providing patients with the risks of insufficient sleep and the benefits of quality sleep may motivate patients to want to change their lifestyles, it still may not be enough to help them make the change. Hershner and O’Brien looked at the effects of an online sleep education program, Sleep to Stay Awake, on college students. They found that this online sleep education intervention did effectively improve knowledge on sleep, and more importantly, led to improved sleep behaviors and quality [227]. Sleep education programs are also effective for seniors. The Sleep Education for Elders Program (SLEEP) led to less poor sleep hygiene behaviors and decreased levels of daytime sleepiness [228]. It is important to provide patients with information about the importance of quality of sleep, but another key piece of this puzzle is to provide access to sleep education materials so they can learn about strategies to help improve their sleep quality.

Another avenue worth exploring is the KAP survey model (knowledge, attitude, practice), which looks at qualitative and quantitative data to help administrators find misconceptions and/or misunderstandings in order to help them promote the changes they desire. KAP has been applied in many areas of medicine. For example, one study looked at how health education could improve KAP in patients with type II diabetes mellitus and subsequently improve clinical glycated hemoglobin (HbA1C) levels. They found that the group that received health education had significantly higher KAP scores from baseline and significantly lower HbA1C compared with controls who were not given health education [229]. Physicians need to prioritize learning how to effectively communicate with their patients about topics like sleep health in order to best find how to support them in making lifestyle changes. In order to do this, physicians must learn more about the importance of sleep during their medical education. Al-Naggar et al., using KAP surveys, found that knowledge about sleep health among medical students was generally low, indicating that there needs to be educational strategies to improve understanding of this topic [230]. Thus, physicians need to learn more about sleep medicine during their education, as well as how to effectively communicate its importance with patients.

Additionally, physicians should engage with their patients to learn more about their unique lives and interests. For many individuals, the general guidelines to motivate lifestyle changes are not always enough and can even be meaningless. If patients could learn about how improving their sleep health would facilitate their ability to do the things they love or accomplish their unique goals, this could be the calling that triggers the change. These motivators vary for different individuals; for some it may be wanting to treat their disease, boosting their focus to teach or learn better, maximizing their job efficiency, enjoying everyday life activities, and/or the potential to improve a mental health struggle they have battled for years. For most people, there is more than one goal and motivators that can be ignited or boosted. Physicians must try to find what uniquely motivates each patient and try to individualize their advice.

## 10. Conclusions

When considering how modern society can manage the growing prevalence of chronic diseases like obesity, depression, and diabetes, inadequate sleep health is a key factor to identify and change. Short/long sleep or poor sleep quality increase the risks of numerous chronic diseases, and as seen throughout this paper, the relationships between sleep insufficiency and these different conditions are all interconnected (Figure 17). All of these diseases together create a massive debt in terms of both dollars and lives. Providing physicians with the tools to help educate their patients on sleep health is an important step in helping to facilitate lifestyle changes By targeting sleep as a lifestyle modification, we can reduce the risks of so many life threatening and life altering diseases. Beyond medical diagnoses, having a healthy sleep schedule is also a good way to improve self-perceived quality of life. In the busy life of the modern world, it might not be easy to get in bed earlier to add an extra hour of sleep, but the risks associated with chronic sleep insufficiency in terms of both health and quality of life are too high to not make this effort.

## Figures and Tables

**Figure 1 ijerph-19-10740-f001:**
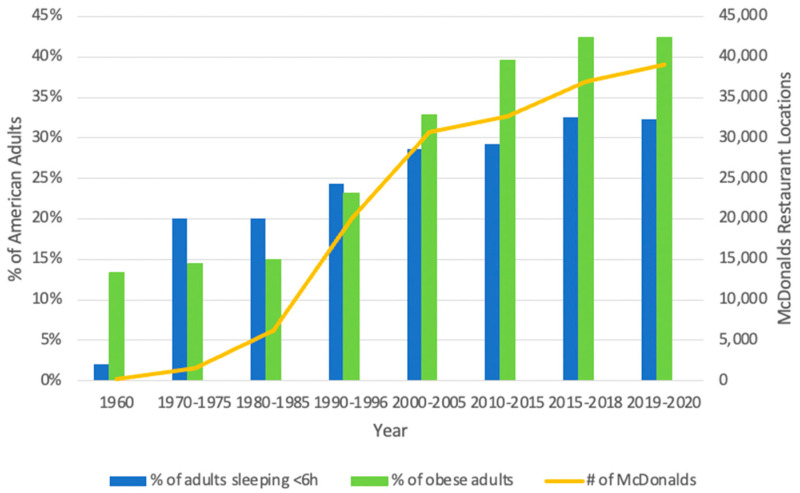
Trends in sleep debt, obesity, and fast food since 1960.

**Figure 2 ijerph-19-10740-f002:**
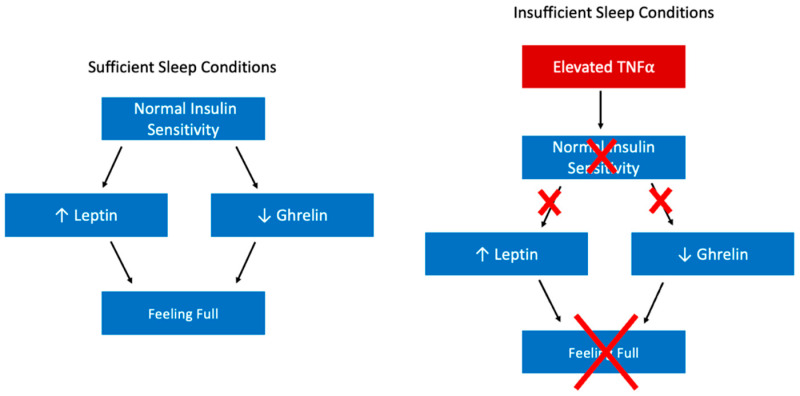
Model for the Effects of Insufficient Sleep on Satiety. An upward facing arrow indicates an increased level of the index and a downward facing arrow indicates a decreased level of the index. Abbreviation: Tumor necrosis factor (TNF).

**Figure 3 ijerph-19-10740-f003:**
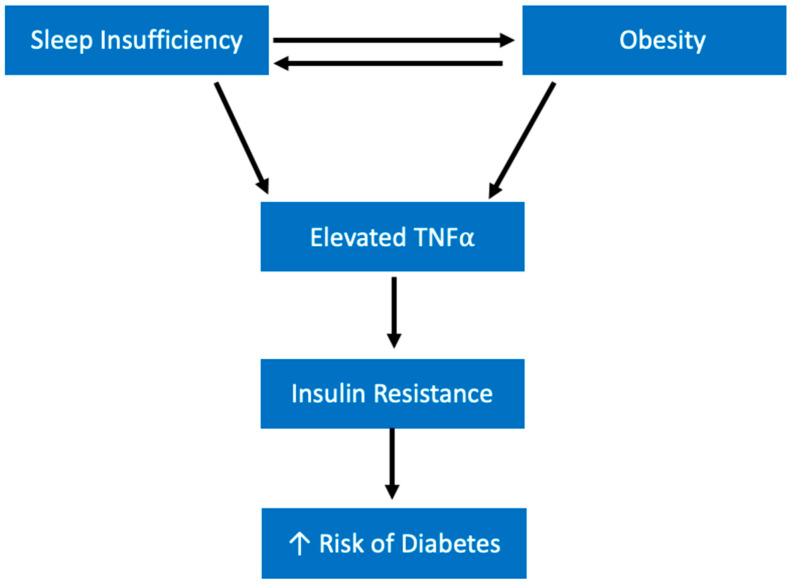
Model for Sleep Loss Effects on Insulin Resistance. An upward facing arrow indicates an increased level of the index. Abbreviation: tumor necrosis factor (TNF).

**Figure 4 ijerph-19-10740-f004:**
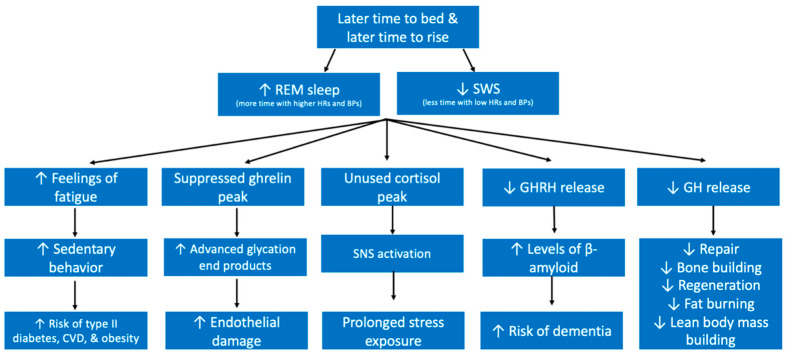
Model of how Evening Chronotype Contributes to Chronic Diseases. Upward arrows indicate increased levels of the index and downward arrows indicate decreased levels of the index. Abbreviations: rapid eye movement (REM), slow wave sleep (SWS), growth hormone (GH), growth hormone releasing hormone (GHRH), cardiovascular disease (CVD).

**Figure 5 ijerph-19-10740-f005:**
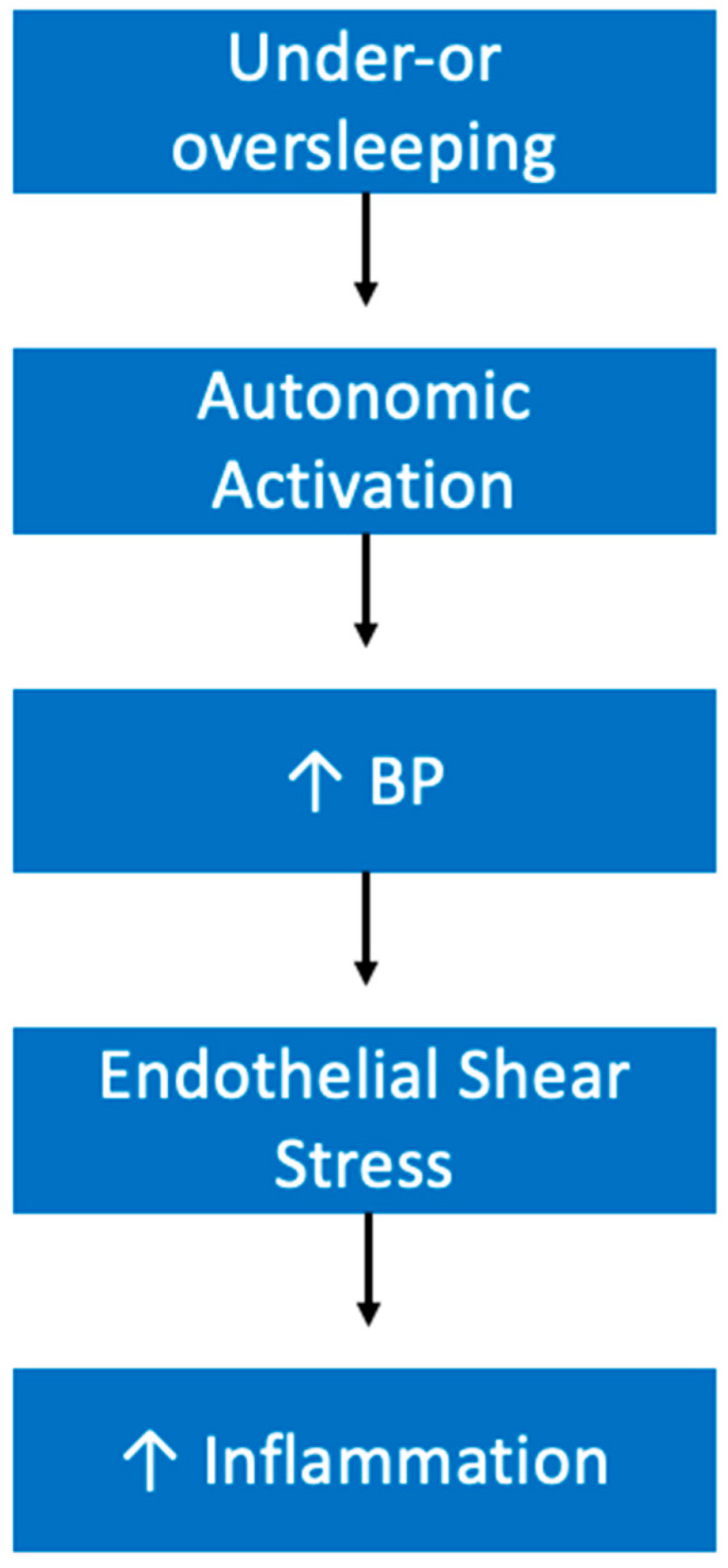
Proposed Mechanism by which Sleep Insufficiency Increases Inflammation. Upward facing arrows indicate increased levels of the index. Abbreviation: blood pressure (BP).

**Figure 6 ijerph-19-10740-f006:**
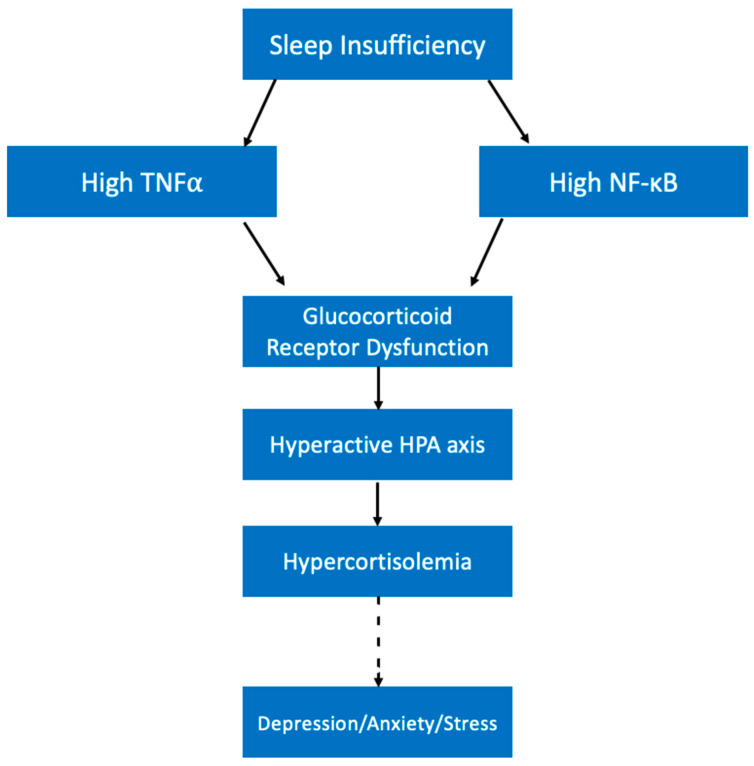
Model for the Influence of Sleep Insufficiency on Mental Distress. Abbreviations: hypothalamic-pituitary-adrenal (HPA), tumor necrosis factor (TNF), nuclear factor (NF).

**Figure 7 ijerph-19-10740-f007:**
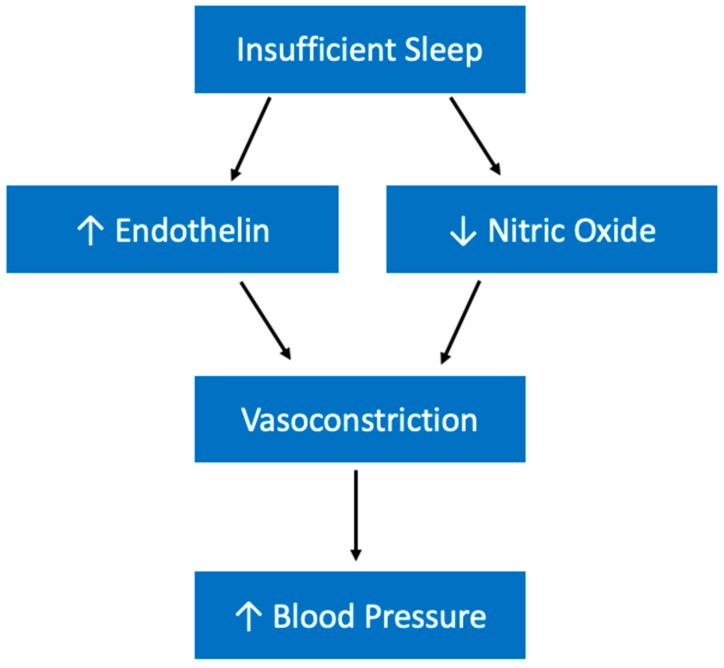
Model of proposed relationship between sleep insufficiency and hypertension. Upward facing arrow indicates increased levels of the index and downward facing arrow indicates decreased levels of the index.

**Figure 8 ijerph-19-10740-f008:**
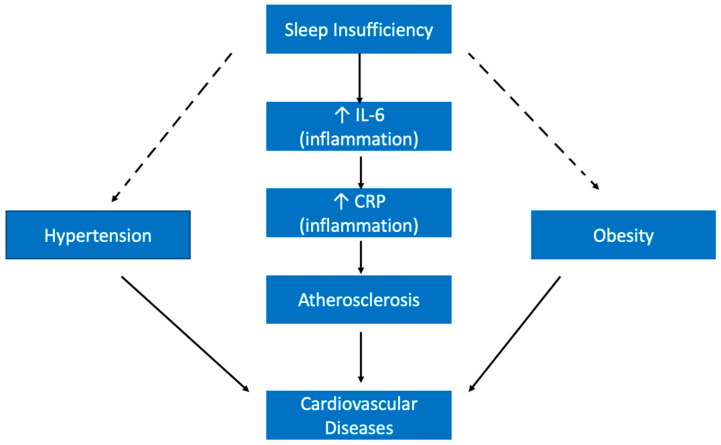
Model of Proposed Relationship between Sleep Insufficiency and Cardiovascular Diseases. Upward facing arrows indicate increased levels of the index. Abbreviation: C-reactive protein (CRP), interleukin (IL).

**Figure 9 ijerph-19-10740-f009:**
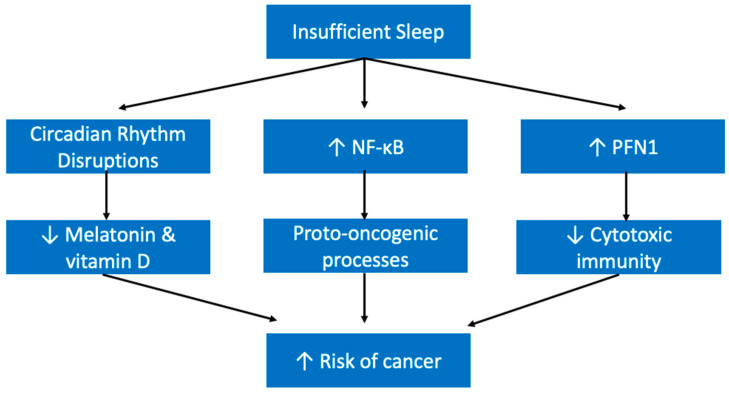
Proposed Model of Relationship Between Inadequate Sleep Health and Cancer. Upward facing arrows indicate higher levels of the index and downward facing arrows indicate lower levels of the index. Abbreviation: profillin-1 (PFN1), nuclear factor (NF).

**Figure 10 ijerph-19-10740-f010:**
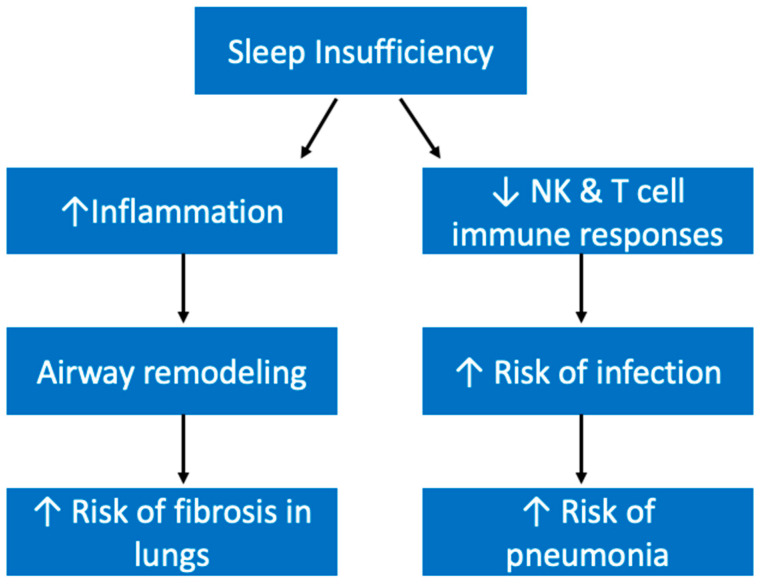
Proposed Model for Relationship Between Insufficient Sleep and Pulmonary Diseases. Upward facing arrows indicate increased levels of the index and downward facing arrows indicate decreased levels of the index. Abbreviations: Natural killer (NK) cell.

**Figure 11 ijerph-19-10740-f011:**
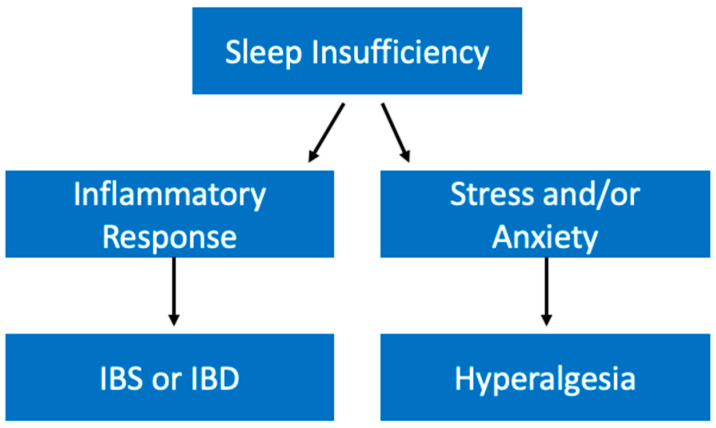
Proposed Model for Relationship Between Insufficient Sleep and Gastrointestinal Diseases. Abbreviations: Irritable bowel syndrome (IBS), inflammatory bowel disease (IBD).

**Figure 12 ijerph-19-10740-f012:**
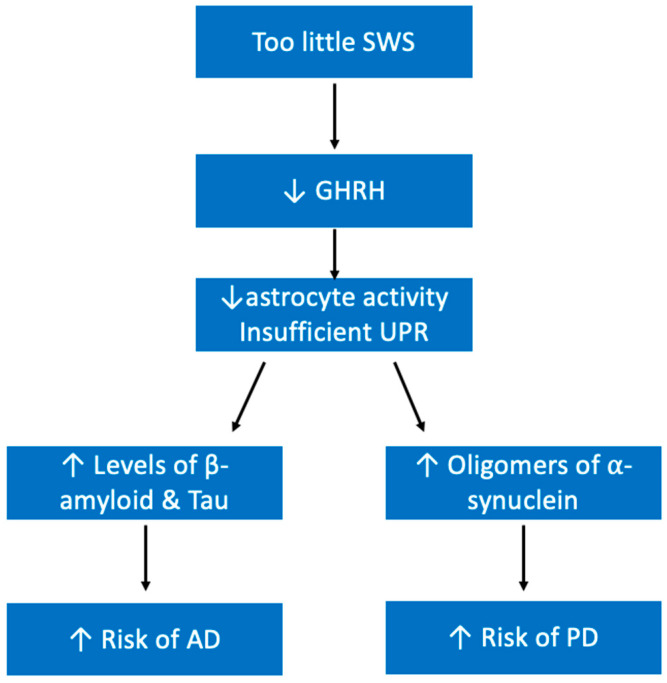
Proposed Model of Relationship between Insufficient Sleep and Neurological Diseases. Upward facing arrows indicate increased levels of the index and downward facing arrows indicate decreased levels of the index. Abbreviations: Alzheimer’s Disease (AD), Parkinson’s’ Disease (PD), unfolded protein response (UPR), slow wave sleep (SWS).

**Figure 13 ijerph-19-10740-f013:**
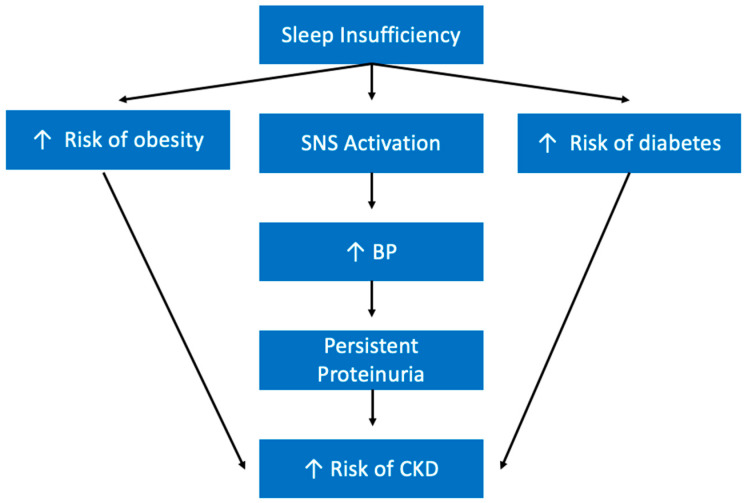
Proposed Model for Relationship Between Sleep Insufficiency and Renal Diseases. Upward facing arrows indicate increased levels of the index. Abbreviations: blood pressure (BP), chronic kidney disease (CKD).

**Figure 14 ijerph-19-10740-f014:**
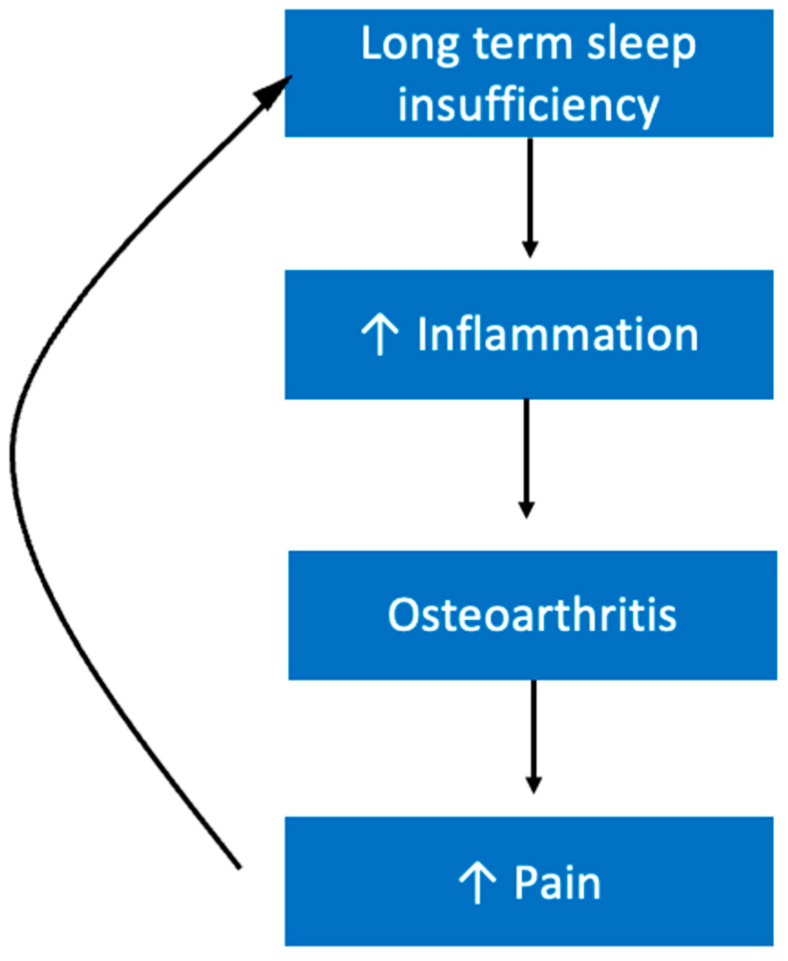
Proposed Model for Relationship between Sleep Insufficiency and Pain. Upward facing arrows indicate increased level of the index.

**Figure 15 ijerph-19-10740-f015:**
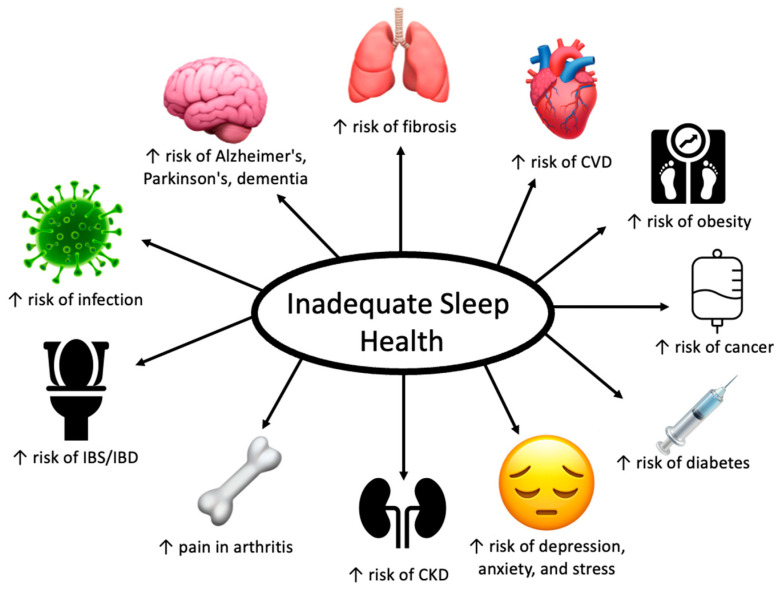
Comprehensive figure illustrating the number of conditions inadequate sleep health can contribute to. Upward arrows indicate increased levels of the index.

**Figure 16 ijerph-19-10740-f016:**
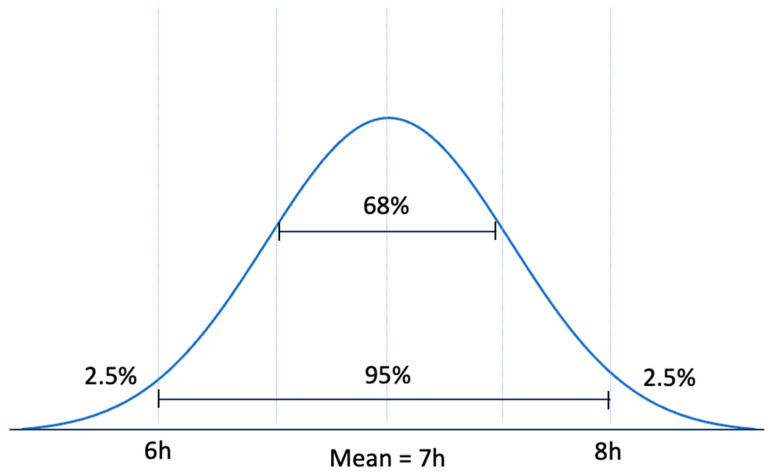
Standard Bell Curve representing that 95% of the population fall within two SDs of the mean.

**Figure 17 ijerph-19-10740-f017:**
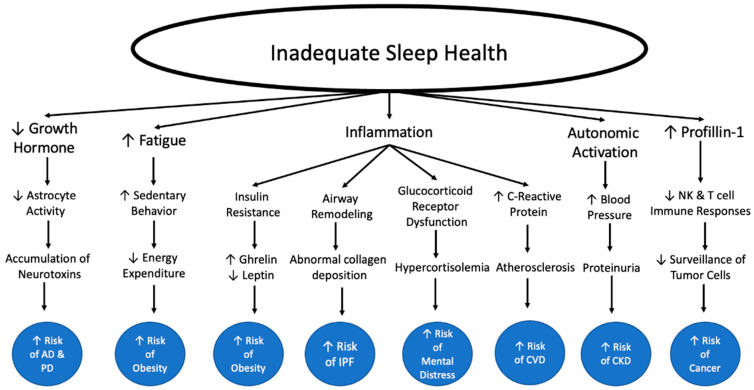
Comprehensive Figure Detailing Mechanisms Connecting Inadequate Sleep Health to Chronic Diseases. Upward facing arrows indicate increased level of the index Abbreviations: Alzheimer’s Disease (AD), Parkinson’s Disease (PD), Idiopathic Pulmonary Fibrosis (IPF), Cardiovascular Disease (CVD), Chronic Kidney Disease (CKD).

## Data Availability

Not applicable.

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
