# Peer review of "Healthy Sleep Every Day Keeps the Doctor Away"

_ijerph, 2022, doi:10.3390/ijerph191710740_

Round 1
Reviewer 1 Report
The paper is written well but the shortfall is that the authors did not indicate whether the paper is general literature and a scoping review. This is something that was noted in the paper. If the authors could provide the information about the methodological process used in the paper, it will enhance the insight of the readers of the journal.

Reviewer 2 Report
This review discusses sleep insufficiency (duration, quality) in relation to a wide range of chronic conditions, including obesity, metabolic syndrome, diabetes, hypertension, cancers, pulmonary diseases, gastrointestinal diseases, neurological diseases, renal diseases, and musculoskeletal diseases. Throughout the manuscript, the authors have synthesized a set of pathways connecting sleep to these chronic diseases which I found very informative. Overall, the work done is exhaustive and complete. It is with great pleasure that I reviewed this article and I congratulate the authors for their efforts. However, as explained below, I raised some recommendations and suggestions that need to be addressed
- Giving that the authors have not only covered sleep duration but also sleep quality and schedules, I recommend them to refer to the “Sleep Health” concept (which includes all of these features) instead of “sleep insufficiency”. I suggest briefly introducing Sleep Health at the beginning of the paper. Please find below some relevant references.
Buysse, D. J. (2014). Sleep health: can we define it? Does it matter? Sleep, 37(1), 9-17.
Hale, L., Troxel, W., & Buysse, D. J. (2020). Sleep health: an opportunity for public health to address health equity. Annual review of public health, 41, 81.
- The second point that needs consideration relate to the general structure of the paper as it stands:
1. Introduction
2. Is insufficient sleep contributing to the growing prevalence of chronic conditions?
3. Sleep insufficiency and obesity
4. Sleep insufficiency and depression/anxiety/stress
4.1. Sleep Schedule and Metabolic Syndrome
5. Contributions of sleep insufficiency on other chronic diseases
5.1. Inflammation
5.2. Diabetes Mellitus
5.3. Hypertension
5.4. Other Cardiovascular Diseases
5.5. Cancers
5.6. Pulmonary Diseases
5.7. Gastrointestinal Diseases
5.8. Neurological Diseases
5.9. Renal Diseases
5.10. Musculoskeletal Diseases
5.11. An Additional Consideration
5.12. Sleep Schedule Variations
6. Benefits of Quality of Sleep on Chronic Diseases
7. Measuring Sleep Quality
8. Sleep Education
9. Conclusion
I’m confused with the presentation of the section “4.1. Sleep Schedule and Metabolic Syndrome” under “4. Sleep insufficiency and depression/anxiety/stress”. This also applies to “5.1. Inflammation”: to my opinion, it might be more appropriate to include all chronic diseases together in the same section. However, inflammation could not be considered a chronic disease but rather an immune process. It looks to me more coherent to address this point prior to this section. Section “2. Is insufficient sleep contributing to the growing prevalence of chronic conditions?” does not treat the announced question and could be included in “3. Sleep insufficiency and obesity. I also suggest addressing metabolic syndrome right after the obesity section giving the link between the two conditions.
- Figure 1 illustrates the relationship between changes in sleep duration and increases in obesity prevalence, but I found the display of the number of McDonald's restaurants to be rather anecdotal. I disagree that this is representative of behavioral changes over time since no direct relationship was established or referenced here. In my opinion, representing data on physical activity, sedentary behavior or energy intake will be more appropriate.
- “Section 3. Sleep insufficiency and obesity”
- This section could benefit from further information and references (below) on how sleep duration and quality affect energy balance. Some statements must be nuanced as the effect on energy expenditure remains uncertain. This also applies to the effect of sleep on the leptin/ghrelin ratio.
Chaput, J. P. (2014). Sleep patterns, diet quality and energy balance. Physiology & behavior, 134, 86-91.
St-Onge, M. P. (2013). The role of sleep duration in the regulation of energy balance: effects on energy intakes and expenditure. Journal of Clinical Sleep Medicine, 9(1), 73-80.
- Aside from ghrelin and leptin the effect of sleep on GLP-1 and peptide YY have to be brought up.
- More emphasis should be also accorded to the non-hormonal effects of sleep on food intake, especially when evoking reward-processing brain regions (line 187). The authors must add Stress/HPA axis activation, modulation of endocannabinoid signaling, and their effects on inhibitory executive function, and impulse control during sleep deprivation. Sleep staging mainly the role of REM sleep in the regulation of appetite could be briefly stated. Please find here some relevant references.
Demos, K. E., Sweet, L. H., Hart, C. N., McCaffery, J. M., Williams, S. E., Mailloux, K. A., ... & Wing, R. R. (2017). The effects of experimental manipulation of sleep duration on neural response to food cues. Sleep, 40(11), zsx125.
Geiker, N. R. W., Astrup, A., Hjorth, M. F., Sjödin, A., Pijls, L., & Markus, C. R. (2018). Does stress influence sleep patterns, food intake, weight gain, abdominal obesity and weight loss interventions and vice versa?. Obesity Reviews, 19(1), 81-97.
- “Section 4.1. Sleep Schedule and Metabolic Syndrome”.
- I want to bring the attention of the authors that in this section they treated chronotype and metabolic syndrome rather than sleep schedule which actually varies in response to a range of factors (social pressures, shift work…)
- The claims that evening chronotype is associated with more time spent on REM sleep should be reconsidered. First, when time spent in bed is fixed there are no differences in sleep staging between chronotype (please refer to Mongrain et al. (2005, 2007)). Second, in naturalistic conditions evening, chronotypes go to bed later and are pushed by social pressure to wake at the same time as the remaining chronotypes. This put them at increased risk of sleep deprivation and severely penalizes their REM sleep (as the homeostatic pressure of sleep dissipates throughout the beginning of the night).
- I urge authors to look at the link between REM sleep, obesity, and metabolic syndrome. Do individuals with obesity have more or less REM sleep compared to their normal-weight counterparts?
- Line 333-335: REM sleep is also important for memory processing and learning. Reduction of REM sleep is also related to increased risks of dementia
- Overall, in this section, the authors should consider the effect of late sleep schedule and circadian misalignment rather than sleep staging. Moreover, they should also discuss studies focusing on sleep and metabolic syndrome outcomes (insulin resistance, disturbed lipid metabolism …). I suggest replacing some part of the obesity section here (part form line 135 to 217).
- As stated previously “Section 5.1. inflammation” should appear in the metabolic syndrome section.
Round 2
Reviewer 2 Report
I want to thank the authors for their quick feedback. I believe that they raised a number of my concerns. However, please see below for further remarks:
Although I agree that section: 2.3 Interaction Among Sleep, Nutrition, and Physical Activity (line 261-294) could be interesting, I'm afraid that focusing on the athlete model would be confusing here, especially as this specific population present concerning sleep outcomes. The authors should understand that elite sport takes a larger dimension than physical activity and oppose athletes to a bench of constraints that may affect sleep health and not always in a positive way. Given that this subject is extensive and may fall out of the scope of this review, I suggest focusing on the main question of this section. In other words, how does physical activity affect sleep health? and how does nutrition affect sleep health? There is already a whole body of recent literature highlighting these interactions in the general population or those with obesity. Some studies even suggested the existence of a virtuous circle: having a good eating pattern along with an active lifestyle would lead to better sleep health and vice-versa.
Please find below some helpful resources:
Zuraikat, F. M., Wood, R. A., Barragán, R., & St-Onge, M. P. (2021). Sleep and diet: mounting evidence of a cyclical relationship. Annual Review of Nutrition, 41, 309-332.
Kredlow, M. A., Capozzoli, M. C., Hearon, B. A., Calkins, A. W., & Otto, M. W. (2015). The effects of physical activity on sleep: a meta-analytic review. Journal of behavioral medicine, 38(3), 427-449.
Saidi, O., Rochette, E., Bovet, M., Merlin, E., & Duché, P. (2020). Acute intense exercise improves sleep and decreases next morning consumption of energy‐dense food in adolescent girls with obesity and evening chronotype. Pediatric Obesity, 15(6), e12613.
Another section that needs substantial amendments is "3. Sleep Chronotype and Metabolic Syndrome":
The authors should note that poor sleep quality in relation to metabolic syndrome is mainly mediated by increased sleep fragmentation through arousals. This generally leads to increased lighter sleep stages. Conversely, both SWS and REM sleep are essential for health. N3+REM/TST (%)> 50% was even considered a marker of sleep quality and continuity.
Sleep fragmentation impairs several hormones secreted during SWS (for instance GH), increases sympathetic excitation, systemic inflammation, oxidative stress, and leads to metabolic dysregulation through insulin resistance and dyslipidemia. To my opinion, this section should focus on these pathways.
The study by pépin et al (2021) documented an increase in REM sleep in late chronotype because of increased time spent in bed (they were not required to wake up early during the COVID pandemic). Early wake-up call generally results in a larger reduction of REM sleep in those getting to bed later than those getting to bed early. As explained in my previous report, no significant differences were found in sleep staging between chronotypes when time in bed is fixed (which is not the case in the study by Harfmann et al. (2020).
The speculation that people who get to bed later have a larger amount of REM sleep compared to those getting earlier here cannot be applicable to real-life settings and cannot explain metabolic disruption. Thus, I recommend correcting this point before moving forward. Moreover, some sentences such as (line 360-361) "It is thought that this is because sleeping metabolic rate is highest during REM sleep, therefore a reduction may contribute to a skewed energy balance" are wrong giving the minimal energy conservation rate during sleep.
However, in addition to sleep fragmentation, another missed pathway here is the circadian misalignment effect on metabolic syndrome. Here I underline that later chronotype is at higher risk of misalignment.
In the light of all these remarks please amend this section as well as Figure 4
